# Physical activity in women attending a dissonance-based intervention after Roux-en-Y Gastric Bypass: A 2-year follow-up of a randomized controlled trial

Sofie Possmark[1], Fanny Sellberg[1], Ata Ghaderi[2], Per Tynelius[1,3], Mikaela Willmer[4], Finn Rasmussen[1], Margareta Persson[5], Daniel Berglind[1,3]*

1 Department of Global Public Health, Karolinska Institutet, Stockholm, Sweden, 2 Department of Clinical Neuroscience, Karolinska Institutet, Stockholm, Sweden, 3 Centre for Epidemiology and Community Medicine, Stockholm County Council, Stockholm, Sweden, 4 Department of Caring Sciences, University of Gävle, Gävle, Sweden, 5 Department of Nursing, Umeå University, Umeå, Sweden

* Daniel.berglind@ki.se

**Data Availability Statement:** Swedish secrecy law prohibits us from making register data publicly available. The data supporting our findings were used under license and ethical approval for the

## Abstract

### Background

The majority of Roux-en-Y gastric bypass (RYGB) patients are not sufficiently physically active post-surgery, yet little support from the Swedish healthcare system is offered. We investigated if a dissonance-based group intervention, aiming to increase health-related quality of life after surgery, had any effect on patients' physical activity two years post-RYGB.

### Methods

Women undergoing RYGB surgery were recruited from five Swedish hospitals and randomized to intervention or control group (standard post-surgery care). The dissonance-based intervention was conducted three months post-RYGB and consisted of four group sessions, each with a specific topic, of which one addressed physical activity. ActiGraph GT3X+ accelerometers were used to measure physical activity at pre-RYGB, one- and two-years post-surgery.

### Results

At pre-RYGB, 259 women were recruited and randomized (intervention n = 156 and control n = 103). Participants had a mean age of 44.7 years (SD 10.3) and pre-RYGB body mass index of 40.8 (SD 4.5) kg/m$^2$. At two-years follow-up, 99 participants (63.5%) in intervention group and 68 (66.0%) in control group had valid accelerometer-measurements. Pre- to post-surgery increases were seen in all physical activity outcomes, but no statistically significant differences between the groups were observed at the two-years follow-up, and intervention effects were poor (d = 0.02–0.35).

current study. Readers interested in obtaining microdata or replicating our study may seek similar approvals and inquire through Statistics Sweden. For further advice see: https://www.scb.se/en/services/guidance-for-researchers-and-universities/.

**Funding:** This study was funded by: the Swedish Research Council (Vetenskapsrådet) grant number 2015-02621 to DB and FR (https://www.government.se/government-agencies/the-swedish-research-council-vetenskapsradet/); the Stockholm County Council (ALF Medicine) grant number 20180266 to DB (https://ki.se/en/nvs/about-the-research-school-in-health-science); and the Research School of Caring Sciences at Karolinska Institutet (NVF), grant number 2-14672016 to DB (https://ki.se/en/nvs/about-the-research-school-in-health-science). The funders had no role in study design, data collection and analysis, decision to publish, or preparation of the manuscript.

**Competing interests:** The authors have declared that no competing interests exist.

## Conclusion

To our knowledge, this is the first dissonance-based intervention targeting women undergoing RYGB surgery. At two-years follow-up, we did not observe any differences in physical activity levels between the intervention group and control group.

**Trial registration number**: ISRCTN16417174.

## Introduction

Bariatric surgery has shown significant weight loss with successful long-term weight maintenance [1, 2]. Physical activity, especially moderate-to-vigorous physical activity (MVPA), is important as it can help to maintain post-bariatric surgery weight loss, improve body composition [3–5], increase cardiorespiratory fitness [3], increase muscle strength, and limit the loss of fat-free mass [6, 7]. A systematic review has shown that exercise interventions performed after bariatric surgery resulted in improved physical fitness and optimized fat mass loss, as well as weight loss [8]. In contrast, a more recent systematic review examining the effect of exercise on weight loss concluded that exercise post-surgery does not contribute to greater weight loss compared to the usual post-surgery care. However, this review only examined weight loss (not loss of fat mass per se) as an outcome [9]. Despite all established health benefits associated with physical activity, and global physical activity guidelines recommending ≥150 min of MVPA per week [10, 11], most patients are not sufficiently physically active pre-surgery and continue to be inactive post-surgery despite large weight loss. However, data remains limited on this issue [12–14]. According to a systematic review and meta-analysis, where physical activity was measured objectively post-surgery, patients do not increase their physical activity [12]. In contrast, another meta-analysis has shown significant increases in objectively measured physical activity after six months and up to three years post-bariatric surgery [13]. Exercise interventions could have beneficial effect on physical activity post-surgery [14, 15]; A randomized controlled trial (RCT), investigating if exercise after surgery could increase health-related quality of life (HRQoL), has reported a small increase in step counts, light physical activity (LPA), MVPA and general health in the intervention group, but improvements were not maintained long-term [14]. Also, individualized physical activity counseling may increase physical activity post-bariatric surgery [15].

Dissonance-based interventions are built on cognitive dissonance theory and aim to change a person's behavior by altering their cognitions [16, 17]. A systematic review has shown that the effectiveness of dissonance-based interventions on non-clinical health behavior change (i.e. changes in health behavior, attitude and intention in non-clinical settings, and not for example pathological behavior such as eating disorder symptomatology) seems positive, even though the included studies were prone to bias [18]. Dissonance-based interventions have shown effects in behavioral change; for healthy physical activity behaviors [19], smoking cessation [20] and the prevention of eating disorders and unhealthy weight gain [19–21]. Moreover, the intervention effects were greater when there were more dissonance-inducing activities, larger group sizes and more group sessions [22]. However, the effect of an already existing program to prevent unhealthy weight-gain increased when additional dissonance-based activities were added [23]. To the best of our knowledge, dissonance-based interventions have never previously been conducted in patients undergoing RYGB-surgery.

In Sweden, hospitals performing bariatric surgery procedures have standard post-surgery care that includes health check-ups by physicians, nurses and/or dieticians (check-ups are

approximately scheduled at six months, one-, two- and five-years post-surgery). However, there is no additional psychosocial or behavioral change support offered in standard care. For this reason, together with the knowledge about the dissonance-based interventions' positive effects on several health outcomes [18–23], we hypothesized that such an intervention could also have an effect on various wellbeing-related outcomes after RYGB surgery. We therefore developed a dissonance-based group intervention program targeting RYGB-treated women, aiming to prevent a decline in various long-term wellbeing-related outcomes, with the expectation that physical activity might be positively affected as well. The aim of this study was to examine if there were any differences in objectively-measured physical activity levels, between the intervention group and control group, two years after RYGB-surgery.

## Material and methods

The Wellbeing after Gastric Bypass (WELL-GBP) trial has been approved by the Stockholm Ethical Review Board (registration number: 2013/1847-31/2. Date at which the ethics committee approved the study: December 10[th] 2013). The trial was registered in February 2015 (ISRCTN16417174) and the enrollment started in January 2015. However, the enrollment that was done before the trial was registered only consisted of informing the participating hospitals and to determine the details of the recruitment of participants. No analyses or outcomes have been affected or changed during or after the registration of the trial. The authors confirm that all ongoing and related trials for this drug/intervention are registered. Written consent was obtained from all participants before they entered the trial. A protocol paper with more detailed description of the WELL-GBP trial has been published elsewhere [24], as well as one-year intervention effects [25]. In this paper we analyzed how the intervention affected physical activity two years after RYGB surgery.

### Recruitment and participants

Participants were recruited between January 2015 and June 2017, from five hospitals located in three counties in Sweden. The two-year follow-up measures were finalized in August 2019. At the time of recruitment, RYGB accounted for more than 80% of all bariatric procedures in Sweden, and 75% of the patients were women [26]. Patients were included in the trial if they were female, eligible for RYGB surgery (body mass index (BMI) $\geq$40 kg/m$^2$, or if comorbidities were present: a BMI $\geq$35 kg/m$^2$) and able to understand and speak Swedish. In total, 259 eligible participants provided written consent and were randomized to intervention group (60%, n = 156) or control group (40%, n = 103). Both groups received standard follow-up care from the hospitals. All participants (intervention group and controls) were offered to wear an accelerometer for seven consecutive days, sent to their homes by mail, approximately one-month pre- and one and two years post-RYGB surgery. To wear the accelerometer was optional for the participants. A CONSORT flow-chart of the recruitment and the follow-up measurements is presented in Fig 1.

### Randomization

Approximately two months post-RYGB, participants who had provided informed consent and pre-RYGB data (questionnaires about HRQoL, among others (not included in this study)) were block randomized (in blocks of 5 participants) within their county to either group, by using the SAS 9.4 procedure Proc Plan (SAS Institute Inc., Cary, NC, USA). The random allocation sequence was computer generated into 60% to intervention and 40% to control group. An investigator not involved in the data collection randomized the participants to their allocated group. Blinding of participants or investigators was not possible.

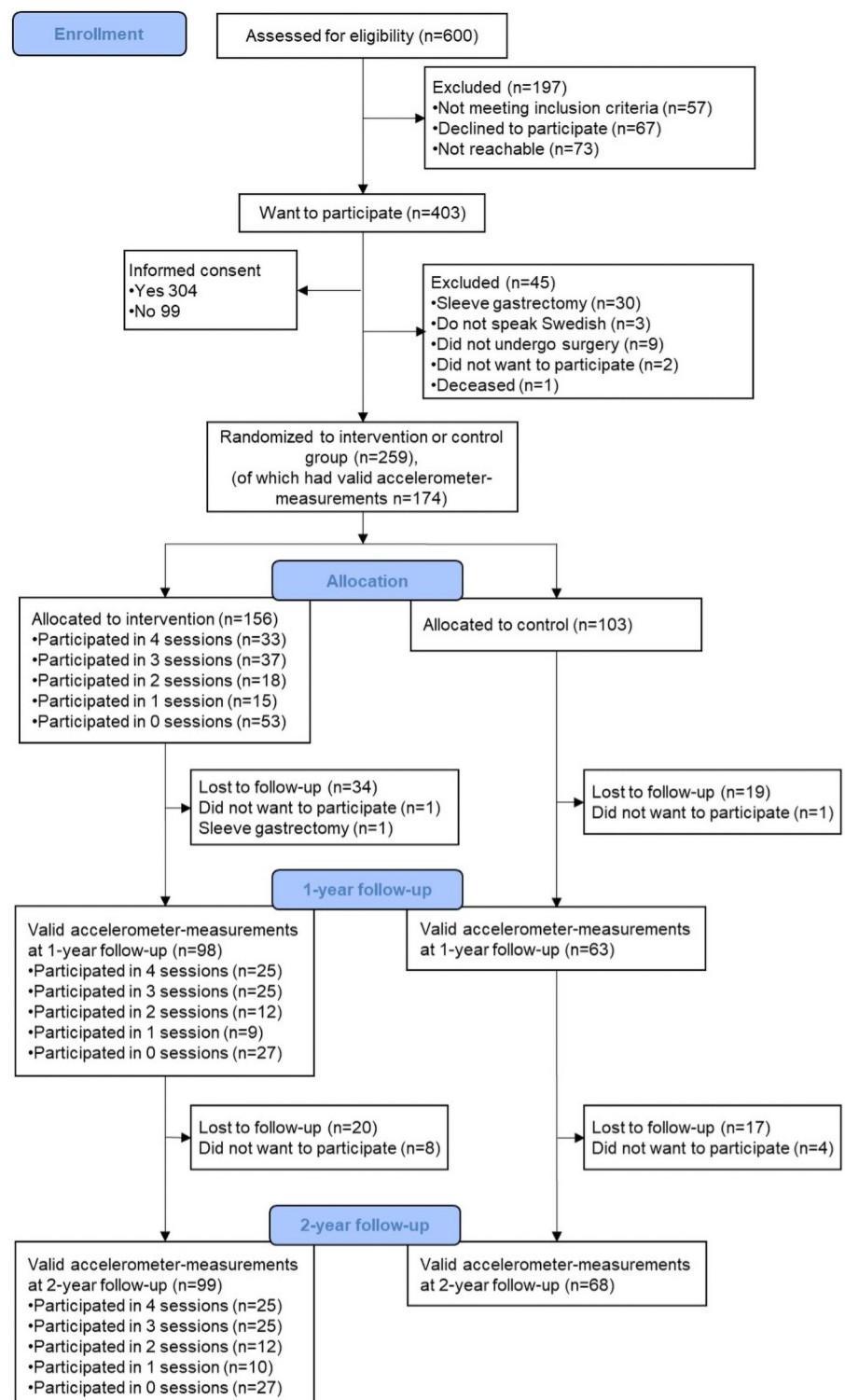

**Fig 1. CONSORT flow-chart.** Participant flow-chart according to CONSORT standards. Of the included participants in the two-year follow-up belonging to the intervention group, 61 of them (61.6%) had attended the session about physical activity. Two participants did not want to participate in the one-year follow-up, but wanted to be contacted for the two-year follow-up.

## Intervention

The intervention in the WELL-GBP trial was based on a model by Stice et al. [20] and consisted of four dissonance-based group sessions post-RYGB. The sessions started approximately three months post-RYGB, were conducted once a week for four weeks, lasted around 1.5 hours/session and were led by a facilitator who had been trained in dissonance-based theory (individual readings on the topic together with informational meetings with a psychologist) and followed an intervention manual. The intervention manual was developed by researchers (AG, DB, MW, FS and FR) who were involved in the development of the intervention. Each session covered a topic known to be problematic for many patients after RYGB surgery: (i) physical activity, (ii) eating behavior, (iii) social relationships and (iv) sexual and intimate relationships. Within each topic, the participants discussed how one should think and act in order to be able to meet future challenges, to optimize health and wellbeing, as well as to maintain the positive outcomes of surgery. The aim of the intervention was to induce dissonance and to increase the probability of healthy actions and attitudes. If a participant had attended at least three out of any of the four sessions, she was considered as having received the intervention. This threshold was set as there is a widely used threshold of <80% adherence to an intervention or medication in order to distinguish adherent from non-adherent patients [27]. Before the intervention started, we conducted a pilot study in order to get feed-back from the pilot participants and to test the intervention manual. Minor changes were done to the manuscript after the feed-back. A description of the pilot study has been published elsewhere [24]. During the session on physical activity, discussions were focused on what sedentary behavior and physical activity is, how one can increase their daily physical activity, difficulties and barriers of physical activity after bariatric surgery and how to overcome these barriers.

## Outcomes

At pre-, one- and two years post-RYGB, all participants were asked to wear the ActiGraph GT3X+ accelerometer on the right hip during all waking hours for seven consecutive days, in order to objectively measure their physical activity levels. Valid accelerometer measurements were set to a minimum wear time of at least 10 hours/day for at least three days. Vector magnitude ($V_m$) was recorded and analyzed in 10-second epochs and converted to counts per minute (cpm). For wear time, an algorithm by Choi et al. [28] was used where non-wear time was classified as non-zero counts for at least 60 minutes, with a maximum break of two minutes. We classified sedentary time as <100 cpm, LPA as 100–3208 cpm and MVPA >3208 cpm [29]. Wear time and classification of bouts were computed using ActiLife v.6.13.3 (ActiGraph, Pensacola, USA).

## Power calculation

The initial power calculations were calculated with HRQoL as main outcome. To attain a statistical power of 0.90 with a significance level of 5% and an expected moderate effect size (Cohen's $d$ = 0.5), estimated participating patients was a sample size of 240, with a 20% expected drop-out rate. Of the original sample of 259, a total number of 167 participants had valid accelerometer measurements at the two-year follow-up (intervention group n = 99, control group n = 68), resulting in a statistical power of more than 90% to detect the pre-planned effect size of d = 0.5.

## Statistical analysis

As one-year differences between the groups already have been published elsewhere [25], this study focused on the two-year follow-up. All participants with valid accelerometer measurements at the two-year follow-up were included in the intention-to-treat analysis, regardless of whether they had valid measurements at pre-RYGB or not. When recruiting participants at pre-RYGB, not all participants had enough time to wear the accelerometer before their surgery, resulting in fewer valid accelerometer measurements at pre-RYGB than at the follow-ups.

Primary analysis was intention-to-treat analysis [30], in order to see if there were any differences in physical activity levels between the intervention and control group at two-years post-RYGB surgery. Normal distribution was evaluated graphically and tested with Shapiro-Wilk test. The variables were approximately normally distributed, however not perfectly. We therefore present the results as means and use Cohen's *d* to quantify intervention effect sizes [31], but because of the lack of complete normality, we also used Kruskal-Wallis H test to test for group differences, which assuming the same distribution in both groups tests whether the medians differ. Also, additional analysis of all outcomes at the two-years follow-up in the intention-to-treat analysis were adjusted for wear time. Chi-square test was used for dichotomous variables, for example if a participant smoked or not, had university level of education or not, and meeting the physical activity recommendations or not. BMI was calculated as weight (kg)/height (m)$^2$.

Furthermore, per protocol analysis was performed to compare the control group to the participants in intervention group who had received the intervention according to protocol (attended ≥3 group sessions). Per protocol analysis was also performed to compare the participants in the intervention group who had participated in the session about physical activity, to the participants who had not participated in that session. Sensitivity analysis was conducted for the participants with valid accelerometer measurements from all three time points, conducted with a regression analysis adjusted for pre-RYGB measures and presented as differences of the means between the groups at one- and two-years follow-up. Additionally, a sensitivity analysis was conducted for all participants who had ≥5 valid days of accelerometer measurements. All statistical analyses were performed using Stata 14.1 (StataCorp) software.

## Results

### Descriptive statistics analysis

A total of 259 women were recruited to the trial, with a mean pre-surgery BMI of 40.8 (SD 4.5) and a mean age of 44.7 years (SD 10.3) at pre-RYGB. Pre-surgery, 54% met the recommended physical activity guidelines of ≥150 minutes of MVPA per week (non-bouts) [10, 11]. At pre-RYGB, 174 of the initial 259 participants had valid accelerometer measurements. However, not all of them where included in the analysis of this study, as only participants with valid measurements at the two-year follow-up were included, regardless if they had valid measurements at pre-RYGB or not (Fig 1). At the two-year follow-up, 167 of the 259 participants had valid accelerometer measurements (intervention group n = 99 and control group n = 68), thus resulting in a loss to follow-up rate of 35.5% (36.5% in intervention group and 34.0% in control group). Of the participants in the intervention group, 61.6% (n = 61) had attended the session where physical activity had been discussed. The only pre-RYGB characteristic that were significantly different between those included in the two-year follow-up (n = 167) compared to those not included (n = 92) were age at surgery (*p* = 0.0097). The women who were included were somewhat older (45.5 years, SD = 10.1) compared to the women not included in this

**Table 1. Pre-surgery characteristics of total sample, intervention group and control group of the women undergoing Roux-en-Y Gastric Bypass (RYGB) surgery and included in the two-year follow-up, as well as the original cohort.**

| Variables | Original cohort (n = 259) | Cohort included in 2-year follow-up (n = 167) | Intervention group (n = 99) | Control group (n = 68) | p-value |
|---|---|---|---|---|---|
| BMI pre-RYGB (kg/m$^2$) (SD) | 40.9 (4.8) | 40.8 (4.4) | 40.8 (4.1) | 41.0 (4.7) | .859 |
| BMI (kg/m$^2$) at 2y post-surgery (SD) | 27.2 (4.1) | 27.0 (3.8) | 26.6 (3.5) | 27.6 (4.2) | .070 |
| % total body weight loss (SD) (from pre- to 2y post-RYGB) | 33.2 (8.8) | 33.6 (8.7) | 34.5 (8.4) | 32.2 (9.0) | .184 |
| % excess BMI loss (excess BMI > 25 kg/m$^2$) (SD) (from pre- to 2y post-RYGB) | 88.1 (23.7) | 89.1 (22.9) | 91.1 (21.1) | 86.1 (25.3) | .103 |
| Age pre-RYGB, in years (SD) | 44.0 (10.4) | 45.5 (10.1) | 44.7 (10.4) | 46.7 (9.5) | .148 |
| Education pre-RYGB, university level, n (%) | 75 (29.1) | 51 (30.5) | 32 (32.3) | 19 (27.9) | .546 |
| Smokers pre-RYGB, n (%) | 17 (6.6) | 9 (5.4) | 4 (4.0) | 5 (7.4) | .352 |

BMI = body mass index.

Presented as mean (standard deviation) or percentage (numbers).

study (41.4 years, SD = 10.6). Table 1 shows pre-RYGB characteristics of the original cohort (n = 259) and of the participants included in this study. No significant differences in pre-RYGB characteristics between intervention group and control group were detected. Seventy-three percent of included participants in the intervention group (of the n = 99 who had valid accelerometer measurements at the two-year follow-up) had attended at least one group session (any of the four sessions), and 51% had received the intervention according to protocol (i.e. attended ≥3 group sessions). There were no statistical differences in pre-RYGB characteristics between those receiving the intervention compared to both control group and participants in the intervention group who did not receive the intervention.

## Intention-to-treat analysis

We observed no statistically significant differences in any of the physical activity levels between intervention and control group at the two-year follow-up, and intervention effects were poor (d = 0.02–0.35). Table 2 shows a detailed description of the outcomes of physical activity intensities at pre-RYGB, one- and two-years follow-up for the intervention and control group. Only the participants who had valid accelerometer measurements at the two-year follow-up (intervention group n = 99 and control group n = 68) were included in the analysis. Of those, 77 in the intervention group and 46 in the control group had valid measurements at pre-RYGB, and 85 and 56 in intervention and control group, respectively, had measures at the one-year follow-up. At two-years, the intervention group spent 29.0 min/day (SE = 1.8) in MVPA and the control group 27.1 min/day (SE = 2.5). Participants were sedentary for 493.3 min/day (SE = 12.0) and 458.8 min/day (SE = 12.3) in the intervention group and control group, respectively. Percentage of women who met the recommended physical activity guidelines (non-bouts) were 66.7% in the intervention group and 54.4% in the control group. No difference between groups were statistically significant.

## Per protocol analysis

Per protocol analysis between participants in intervention group who received the intervention (n = 50) and control group (n = 68) showed no statistically significant differences between the groups, except for sedentary time. Those who received the intervention were more sedentary (522.5 min/day, SE = 17.2) than the control group (458.8 min/day, SE = 12.3) (p = 0.002, d =

**Table 2. Intention-to-treat analysis of the different physical activity intensities, measured by the GT3X+ accelerometers and divided by intervention and control group, in women pre-, one- and two-years post-Roux-en-Y Gastric Bypass (RYGB) surgery.**

| Accelerometer outcomes | Pre-RYGB, Intervention (n = 77) | Pre-RYGB, Control (n = 46) | p-value | 1y post-RYGB, Intervention (n = 85) | 1y post-RYGB, Control (n = 56) | p-value | 2y post-RYGB, Intervention (n = 99) | 2y post-RYGB, Control (n = 68) | p-value | Cohen's d (95% CI) |
|---|---|---|---|---|---|---|---|---|---|---|
| Mean wear time, mean hours/d (SE) | 14.3 (0.1) | 14.0 (0.2) | .154 | 15.0 (0.2) | 14.9 (0.2) | .858 | 15.2 (0.2) | 14.7 (0.2) | .195 | .35 (.03 to .66)* |
| Mean counts, mean min/d (SE) | 558.0 (23.6) | 564.1 (28.1) | .714 | 576.1 (20.3) | 596.8 (24.3) | .541 | 570.1 (17.6) | 579.9 (22.9) | .858 | -.05 (-.36 to .25) |
| MVPA, mean min/d (SE) | 26.7 (2.1) | 24.5 (3.1) | .327 | 27.9 (1.9) | 30.0 (3.1) | .936 | 29.0 (1.8) | 27.1 (2.5) | .302 | .10 (-.21 to .41)* |
| LPA, mean min/d (SE) | 356.3 (10.6) | 359.3 (11.1) | .854 | 390.3 (9.3) | 401.6 (10.6) | .351 | 392.4 (9.2) | 394.0 (11.2) | .916 | -.02 (-.33 to .29) |
| Sedentary time, mean min/d (SE) | 476.7 (11.9) | 455.5 (12.2) | .235 | 481.0 (12.7) | 462.9 (13.1) | .362 | 493.3 (12.0) | 458.8 (12.3) | .075 | .31 (-.00 to .62)* |
| Mean steps, mean counts/d (SE) | 6176.6 (282.1) | 5971.5 (397.9) | .548 | 7518.5 (300.9) | 7571.6 (396.3) | .792 | 7700.3 (268.5) | 7387.8 (366.5) | .268 | .11 (-.20 to .42)* |
| Meeting PA-recommendations**, n (%) | 43 (55.8) | 20 (43.5) | .184 | 48 (56.5) | 32 (57.1) | .937 | 66 (66.7) | 37 (54.4) | .110 | .25 (-.06 to .56) |
| Meeting PA-recommendations in ≥10-min bouts**, n (%) | 7 (9.1) | 4 (8.7) | .941 | 12 (14.1) | 9 (16.1) | .750 | 17 (17.2) | 13 (19.1) | .748 | -.05 (-.36 to .26) |

MVPA = moderate-to-vigorous physical activity; LPA = light physical activity; PA = physical activity. Presented as mean scores (standard errors) or frequency (percent) for each subscale, p-value for the difference in medians between the two groups (Kruskal-Wallis H test) at pre-RYGB, one- and two-years post-RYGB surgery. Effect sizes at two years measured with Cohen's d (95% CI).

*Normal distribution assumption rejected according to Shapiro-Wilk test (p<0.05). Participants with valid accelerometer measurements at the two-year follow-up were included in the analysis. There are fewer participants with valid measurements at pre-RYGB than at the follow-ups, because not all participants had enough time to wear the accelerometer before their surgery.

**PA-recommendations: ≥150 minutes of MVPA per week in non-bouts and 10-minute bouts.

-0.58) at two-years post-RYGB (S1 Appendix). However, this difference was also significantly different at pre-RYGB (*p* = 0.039), and therefore this difference may not be relevant from a clinical perspective.

Per protocol analysis of the participants attending the session about physical activity (n = 61) versus participants in the intervention group who did not attend that session (n = 38), showed no significant differences of clinical importance in any of the physical activity outcomes that changed any of the conclusions. There were also no differences in any of the pre-RYGB characteristics between these two groups.

## Sensitivity analysis

All outcomes at two years post-RYGB in the intention-to-treat analysis were adjusted for accelerometer wear time, but there were still no differences between the groups (*p*>0.05) (Table 2 shows the results from the non-adjusted analysis). Moreover, we performed sensitivity analysis for the women who had valid accelerometer measurements from all measurement time points (pre-RYGB, one- and two-year follow-up, n = 68 in intervention group and n = 36 in control group) and adjusted all outcomes at one- and two years for pre-RYGB measurements, but no statistical differences between the intervention and control group were observed (*p*>0.05). Figs 2 and 3 graphically show the mean daily minutes of MVPA and sedentary time from pre-RYGB to two years, for the women who had valid measurements at all time points (not adjusted for pre-RYGB measures). Fig 4 shows the percentage of women with valid measurements at all time points in intervention group (n = 68) and control group (n = 36) who meet current physical activity guidelines of doing at least 150 minutes of MVPA per week in non-bouts.

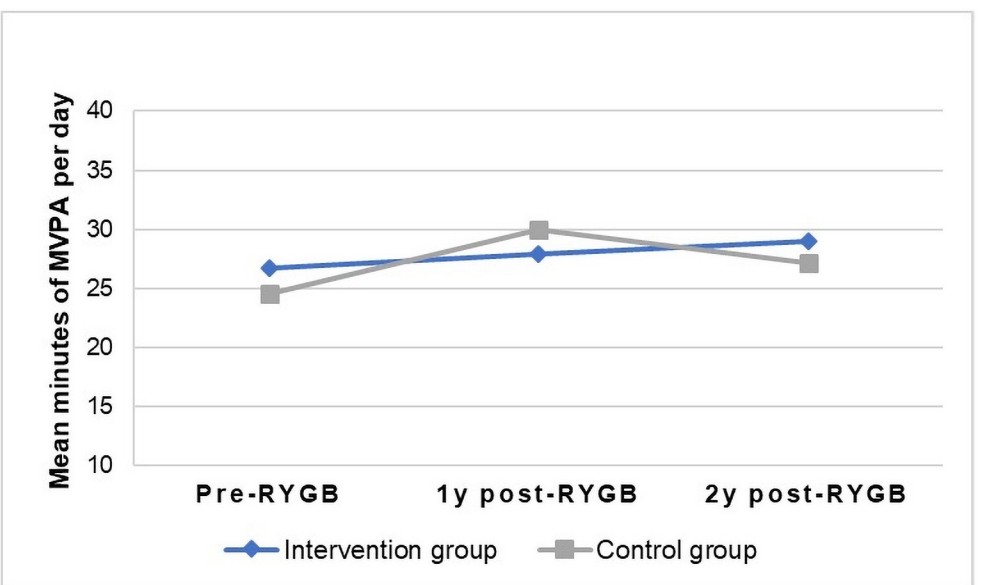

**Fig 2. Differences in daily MVPA over time.** Means of daily minutes of moderate-to-vigorous physical activity (MVPA) with 95% confidence intervals of the women in the intervention group (n = 68) and control group (n = 36) who had valid accelerometer measures at all three assessment time points at pre-, one- and two-years post-Roux-en-Y Gastric Bypass (RYGB) surgery.

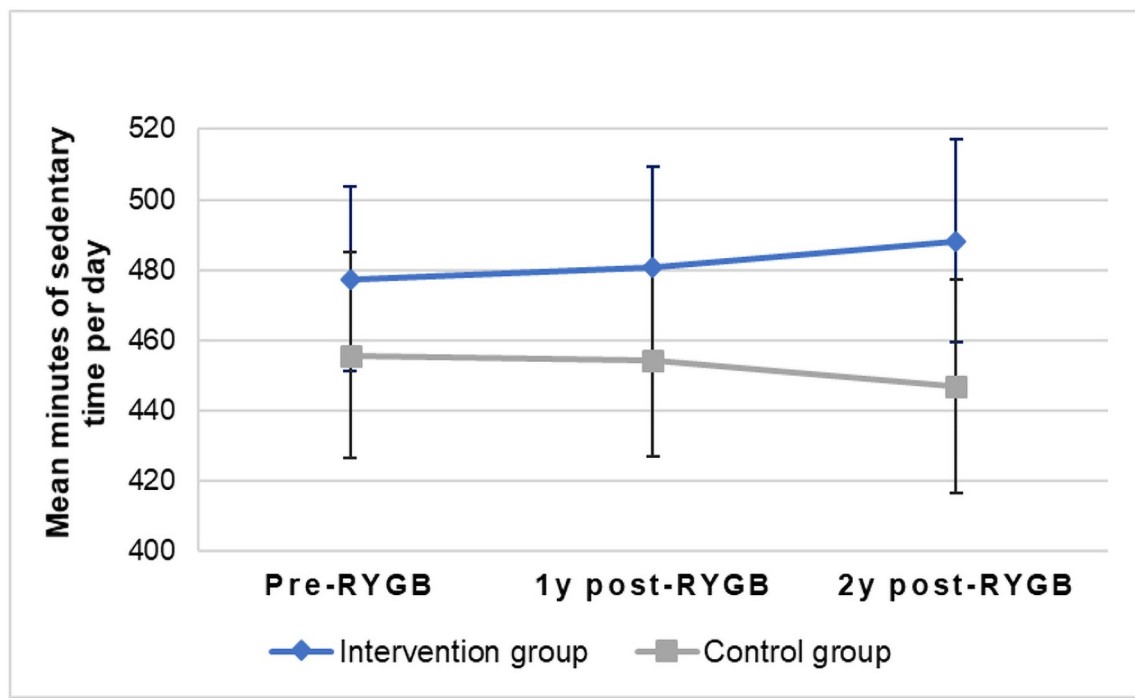

**Fig 3. Differences in daily sedentary time over time.** Means of daily minutes of sedentary time with 95% confidence intervals of the women in the intervention group (n = 68) and control group (n = 36) who had valid accelerometer measures at all three assessment time points at pre-, one- and two-years post-Roux-en-Y Gastric Bypass (RYGB) surgery.

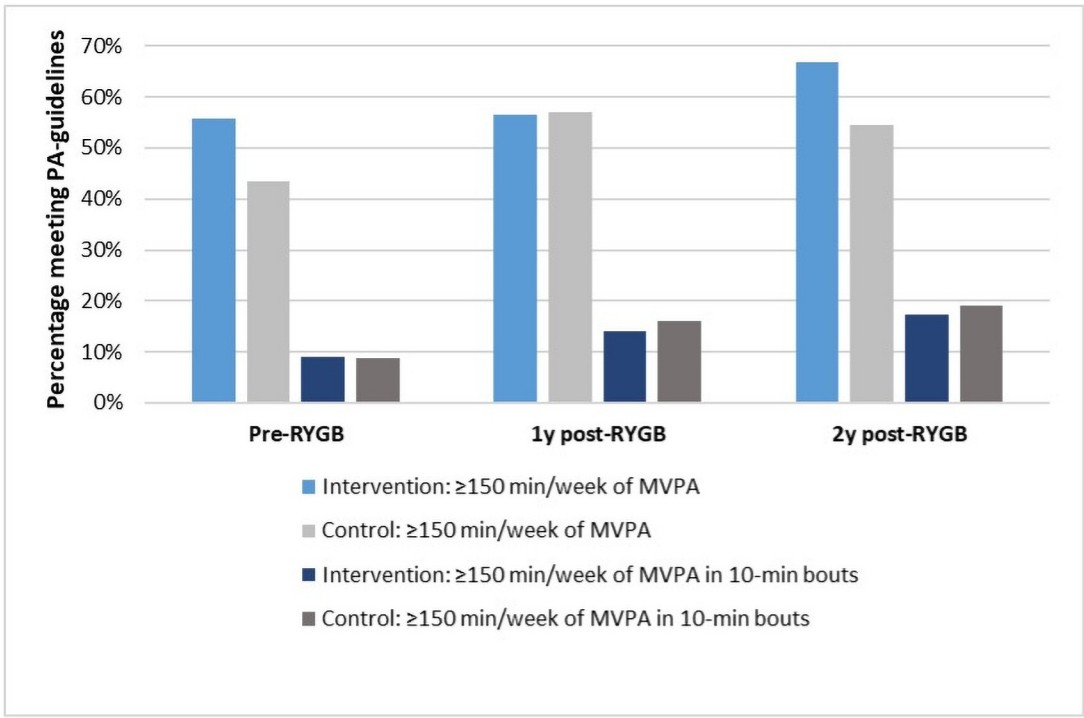

**Fig 4. Percentage meeting physical activity guidelines.** Percentage of women in intervention group (n = 68) and control group (n = 36) who had valid accelerometer measures at all three assessment time points at pre-, one- and two-years post-Roux-en-Y Gastric Bypass (RYGB) surgery who meet current physical activity guidelines of doing at least 150 minutes of moderate-to-vigorous physical activity (MVPA) per week in non-bouts.

We also performed sensitivity analysis for women who had ≥5 valid accelerometer measurement days at pre-RYGB and two-year follow-up (intervention group n = 92 and control group n = 64). No statistically significant between-group differences were observed, except for sedentary time, where the intervention group was significantly more sedentary than the control group ($p$ = 0.047), with a mean of 500.6 min/day (SE = 12.5) compared to 460.8 min/day (SE = 2.8), respectively ($d$ = 0.35) (S2 Appendix).

## Discussion

### Main findings

No statistically significant differences in any of the physical activity outcomes were observed between the dissonance-based intervention group and control group at two years post RYGB.

### Previous research

Some interventions investigating different counselling interventions to improve physical activity before or after bariatric surgery have been conducted [14, 15, 19, 32, 33]. A review from 2013 showed that individualized counselling directed toward physical activity can help bariatric patients to increase their physical activity post-surgery [15]. The interventions included in that review started pre-surgery or between three to 102 months post-surgery and the lengths of the interventions were between six and 12 weeks long [15]. In addition, a previous study has shown an association between meeting the physical activity recommendations of at least 150

min of MVPA per week and a higher score for HRQoL both pre- and one-year post-surgery [34]. A pre-surgery intervention consisting of six counselling sessions with the aim to increase physical activity reported that the intervention group increased their MVPA and also reported better HRQoL scores post-intervention compared with the control group [32]. Further, an intervention where participants were randomized to either standard care, receiving pedometers only, or receiving pedometers as well as counselling sessions, showed that the combined pedometer and counselling group increased physical activity from pre-surgery up to 6.5 months post-surgery (counselling and pedometers = 6 787 steps/day, pedometers only = 5 325 steps/day and standard care = 5253 steps/day) [33]. These results are not in line with our findings, as we did not observe any differences in physical activity between intervention and control groups. However, most of these studies were conducted pre-surgery [32, 33] and with shorter follow-up time and may thus not be quite comparable to our study. For instance, a recent physical activity intervention has shown increases in physical activity levels and step counts at one year post-RYGB, but the effects were not maintained at two-year follow-up [14]; thus, in line with our findings at two-years. However, the main aim of our WELL-GBP intervention was to maintain the increased HRQoL post-RYGB, as HRQoL, for some bariatric patients, starts decreasing around one to two years post-RYGB [35, 36], where we anticipated that physical activity might be positively affected as well. A dissonance-based intervention aiming to increase physical activity among female college students, showed that the participants receiving the dissonance-based intervention had greater increases in physical activity directly at post-test, but not at the follow-up at six months post-intervention [19]. This may indicate that dissonance-based interventions might increase physical activity in the short-term but are not as effective for long-term maintenance of physical activity.

To the best of our knowledge, the WELL-GBP trial is the first dissonance-based intervention for post-RYGB support to women, with the aim to prevent a decline in HRQoL and also possibly increase physical activity. The way the intervention was delivered is of importance to discuss. Of the total participants randomized to intervention and included in this study, 73% attended at least one session (any of the four sessions) and 51% received the intervention according to protocol (Fig 1), despite the fact that all times and dates for the sessions were decided in agreement with the participants. This may indicate that another type of delivery, such as various online-based group sessions, might be more suitable and enable more RYGB patients to attend all sessions. Previous research, conducted on both men and women, have shown that reasons and barriers for not attending support groups after bariatric surgery are due to inconvenient times and locations, responsibilities towards family, work or school, as well as travel time and distance [37]. Also, more group sessions have been observed to be more effective, where four to five sessions seem to be optimal [22]. Therefore, maybe our intervention consisting of four dissonance-based sessions, where only one of them elaborate on physical activity, is too little to show any long-term impact on physical activity behavior post-surgery.

## Strengths and limitations

First, the RCT-design, which is the main strength of the study, enables us to compare the intervention group's results to the standard follow-up care provided by the hospitals. Second, we also pre-registered the trial (ISRCTN16417174) and followed the initial analysis plan. Third, physical activity was objectively measured at pre-RYGB and all follow-ups, with ActiGraph GT3X+ accelerometers, a validated research tool which accurately estimates physical activity in free-living subjects [38]. We chose objective measures as self-reported physical activity among bariatric patients have proven not to be a reliable method [39–41]. Inclusion criterion

for valid wear time was set to at least three days with a minimum of 10-hour wear time per day, as three to four wear time days has shown to be sufficient for achieving 80% reliability of MVPA [42]. Fourth, we based the intervention on Stice's dissonance-based intervention model [20], which has previously shown positive effects on physical activity behaviors [19] and on non-clinical health behavior change [18]. Fifth, the study results have generalizability as we recruited participants from five hospitals from different geographical areas of Sweden, and our participants have similar age and BMI as the typical female RYGB-patient in Sweden [43]. However, the sample of participants included in our study might be somewhat selected, as our participants, compared to the general bariatric surgery patients, may have had higher motivation and interest for health behavior change and therefore be more prone to participate. Thus, when generalizing our results, it is of importance to take this into consideration. Globally, bariatric surgery patients are somewhat older and have higher pre-surgery BMI [44] than the women in this study. Also, the post-surgery care might vary between countries, and therefore our findings should be generalized to other countries with caution. Sixth, few studies have objectively measured bariatric surgery patients' physical activity levels both before and after surgery. The majority of those have only conducted follow-up 6–12 months post-surgery [4, 33, 39–41, 45–47], while some have conducted follow-up assessments after two to four years [14, 48, 49]. Our study therefore adds further knowledge on physical activity patterns among women from before to long-term post-RYGB. Finally, the WELL-GBP trial can easily be implemented in healthcare settings, as various healthcare personnel, such as nurses, dietitians and physiotherapists would be able to deliver the intervention after appropriate training [20].

There are some limitations to the current study that need to be addressed. First, any comparisons or generalizations to men are not possible, as only women were included. We also only included patients who underwent RYGB, therefore any comparisons with other types of bariatric surgeries might be done with some caution. Second, smaller sample sizes were used when conducting per protocol and sensitivity analyses. Consequently, the power might be too small to detect between-group differences. Third, participants were recruited from different hospitals where the follow-up routines differed slightly, which may have influenced the outcomes. Fourth, of the total 259 participants included in the trial at pre-RYGB, 167 (64%) had valid accelerometer measurements at the two-year follow-up, which might have affected the findings as a result of selection bias: i.e. participants who already are more physically active choose to wear an accelerometer, while inactive participants refrained. Though, no differences in pre-RYGB characteristics were observed, except that the 167 women with valid accelerometer measurements were older than those women without valid measurements (45.5 years (SD 10.1) versus 41.4 years (SD 10.6), respectively, $p = 0.0097$). Fifth, only 61.6% of the participants in the intervention group attended the session about physical activity, which might have had impact on the results. Sixth, one limitation of the ActiGraph GT3X+ is that it cannot distinguish between sitting or standing, which might result in inaccurate estimations of sedentary time [50]. Seventh, the power calculation in this RCT was performed to detect group differences at the two years follow-up for the primary outcome (HRQoL) and not secondary outcomes such as physical activity. Finally, the ActiGraph GT3X+ has not been validated in RYGB patients.

## Conclusion

To our best knowledge, this is the first dissonance-based intervention aiming to influence physical activity among women post-RYGB surgery. However, no differences in any of the physical activity outcomes were observed between the intervention group and control group two-years post-RYGB. Dissonance-based interventions focusing exclusively on the topic of

physical activity might be more suitable and should be considered for future dissonance-based interventions aiming to increase physical activity levels among bariatric patients.

## Supporting information

**S1 Checklist. CONSORT 2010 checklist.**
(DOC)

**S1 Appendix. Per protocol analysis of the physical activity intensities two years post-Roux-en-Y Gastric Bypass (RYGB) surgery for the women in the intervention group who received the intervention (attended ≥3 of 4 sessions) versus control group (standard care).**
MVPA = moderate-to-vigorous physical activity; LPA = light physical activity. Presented as mean scores (standard errors) or numbers (percentage) for each variable, p-value for the difference between the groups at the two-year follow-up. Effect sizes at two years measured with Cohen's *d* (95% CI). There are fewer participants with valid measurements at pre-RYGB than at the follow-ups, because not all participants had enough time to wear the accelerometer before their surgery. *PA-recommendations: ≥150 minutes of MVPA per week in non-bouts and 10-minute bouts.
(DOCX)

**S2 Appendix. Sensitivity analysis for pre-surgery and two-years follow-up measures of the physical activity intensities (measured by the GT3X+ accelerometers) among the women, undergoing Roux-en-Y Gastric Bypass (RYGB) surgery, in the intervention group and control group who had ≥5 valid accelerometer measurement days.** MVPA = moderate-to-vigorous physical activity; LPA = light physical activity. Presented as mean scores (standard errors) or numbers (percent) for each subscale, *p*-value for the difference between the two groups at pre-RYGB and two-years post-RYGB surgery. Effect sizes at 2 years measured with Cohen's *d* (95% CI). There are fewer participants with valid measurements at pre-RYGB than at the follow-ups, because not all participants had enough time to wear the accelerometer before their surgery. *PA-recommendations: ≥150 minutes of MVPA per week in non-bouts and 10-minute bouts.
(DOCX)

**S1 File. The protocol study plan for the conduct and analysis of the trial that the ethics committee (the Stockholm Ethical Review Board) approved before the trial began.**
(DOC)

## Acknowledgments

We would like to thank the staff involved in this study from the five hospitals for their help in recruiting study participants and to the study participants who participated in the data collection.

## Author Contributions

**Conceptualization:** Ata Ghaderi, Mikaela Willmer, Finn Rasmussen, Daniel Berglind.

**Data curation:** Per Tynelius.

**Formal analysis:** Sofie Possmark, Per Tynelius.

**Funding acquisition:** Finn Rasmussen, Daniel Berglind.

**Investigation:** Sofie Possmark, Fanny Sellberg.

**Methodology:** Fanny Sellberg, Ata Ghaderi, Mikaela Willmer, Finn Rasmussen, Daniel Berglind.

**Project administration:** Sofie Possmark, Fanny Sellberg.

**Resources:** Finn Rasmussen, Daniel Berglind.

**Software:** Per Tynelius.

**Supervision:** Ata Ghaderi, Finn Rasmussen, Margareta Persson, Daniel Berglind.

**Visualization:** Sofie Possmark, Daniel Berglind.

**Writing – original draft:** Sofie Possmark.

**Writing – review & editing:** Fanny Sellberg, Ata Ghaderi, Per Tynelius, Mikaela Willmer, Finn Rasmussen, Margareta Persson, Daniel Berglind.

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
