## [Decision Letter · Decision Letter 0]

8 May 2020

PONE-D-20-01663

Physical activity in women attending a dissonance-based intervention after Roux-en-Y Gastric Bypass: A 2-year follow-up of a randomized controlled trial

PLOS ONE

Dear Ms Possmark,

Thank you for submitting your manuscript to PLOS ONE. After careful consideration, we feel that it has merit but does not fully meet PLOS ONE’s publication criteria as it currently stands. Therefore, we invite you to submit a revised version of the manuscript that addresses the points raised during the review process.

The manuscript has been evaluated by three reviewers, who provided several comments on reporting and statistical aspects of your study. Please carefully revise your manuscript to address all concerns raised. 

We would appreciate receiving your revised manuscript by Jun 21 2020 11:59PM. To enhance the reproducibility of your results, we recommend that if applicable you deposit your laboratory protocols in protocols.io, where a protocol can be assigned its own identifier (DOI) such that it can be cited independently in the future. For instructions see: http://journals.plos.org/plosone/s/submission-guidelines#loc-laboratory-protocols

We look forward to receiving your revised manuscript.

Kind regards,

Dario Ummarino, Ph.D.

Associate Editor

PLOS ONE

Journal Requirements:

2. Thank you for submitting your clinical trial to PLOS ONE and for providing the name of the registry and the registration number. The information in the registry entry suggests that your trial was registered after patient recruitment began. PLOS ONE strongly encourages authors to register all trials before recruiting the first participant in a study.

a) your reasons for your delay in registering this study (after enrolment of participants started);

b) confirmation that all related trials are registered by stating: “The authors confirm that all ongoing and related trials for this drug/intervention are registered”.

Please also ensure you report the date at which the ethics committee approved the study as well as the complete date range for patient recruitment and follow-up in the Methods section of your manuscript.

Reviewers' comments:

Reviewer's Responses to Questions

**Comments to the Author**

1. Is the manuscript technically sound, and do the data support the conclusions?

Reviewer #1: Yes

Reviewer #2: Yes

Reviewer #3: Yes

2. Has the statistical analysis been performed appropriately and rigorously? 

Reviewer #1: No

Reviewer #2: Yes

Reviewer #3: Yes

3. Have the authors made all data underlying the findings in their manuscript fully available?

Reviewer #1: Yes

Reviewer #2: Yes

Reviewer #3: Yes

4. Is the manuscript presented in an intelligible fashion and written in standard English?

Reviewer #1: Yes

Reviewer #2: Yes

Reviewer #3: Yes

5. Review Comments to the Author

Reviewer #1: General comments

In this paper, authors assessed the effect of a dissonance-based intervention conducted in the first months after RYGB on habitual physical activity up to two years after surgery. They conclude that no difference was observed between physical activity at the two-year follow-up between intervention and control groups. A previous paper by the same group described the effect of the intervention at the 1-year follow-up. Strengths of the study were to use a randomized design and to assess physical activity with accelerometers. The study is well written, and the references are up to date.

My only concern is about the statistical methods used. Authors compared physical activity outcomes between the two groups at each time point (pre-surgery, one-year and two-year post-surgery) instead of comparing longitudinal changes in physical activity. By doing so, they included only participants with valid data at the 2-year follow-up and therefore excluded many participants who had valid data at baseline but not at 2 years. Mixed models might be more appropriate when several time points are available.

Specific comments

Abstract

- Line 28: Is “baseline” a pre-surgery measure? If so, “before surgery” might be more precise than “at baseline”

Introduction

- Line 58-67: the references are up to date but the message of this paragraph is not clear to me. The expression “In addition” (line 65) does not seem appropriate to me, “In contrast” might be more appropriate here.

What I understand from this paragraph is that: 1) exercise training has been shown effective after bariatric surgery, 2) mixed results are found regarding the spontaneous change in physical activity after bariatric surgery, 3) exercise training leads to a small and transient increase in habitual physical activity, 3) physical activity counselling might be effective to increase physical activity.

I would suggest to make a clearer distinction between changes in physical activity related to bariatric surgery and those related to exercise training after bariatric surgery.

- Line 91: “objectively-measured” rather than “objective measured”

Methods

- Line 98-99: “we analysed how the intervention affected physical activity”

Results

- Tables do not fit in the pdf. I cannot comment the tables.

- The authors compared physical activity outcomes between intervention and control group at all times (Table 2). This method results in the exclusion of many participants who do no hate valid measures at the 2-year follow-up. It would have been more appropriate to compare changes in physical activity between groups. Mixed models are usually used in this situation.

Discussion

- Lines 307-309: Are the results similar in other contexts (eg in the studies cited in the introduction)?

Reviewer #2: Thank you for the opportunity to review the manuscript entitled ‘Physical activity in women attending a dissonance-based intervention after Roux-en-Y- Gastric Bypass: A 2-year follow-up of a randomized controlled trial’. This is an interesting paper looking at objectively measured physical activity in a sample of women whom have undergone RYGB, this is a large sample given the population and topic area (physical activity) that the authors are researching in. It is also an area of research which is in its infancy and a useful addition to the literature.

Please see my comments in the document attached

Reviewer #3: Two-year follow-up results on patients activity levels were summarized from a two-arm randomized controlled clinical trial which aimed to increase quality of life following Roux-en-Y gastric bypass surgery through dissonance-based group intervention. At two-years follow-up, no significant differences in physical activity levels were observed between the intervention and control arms.

Minor revisions:

1- Cohen’s d is valid when the distribution of the data is normal. Indicate if the normality assumption was met.

2- Line 172: Technically the Kruskal-Wallis H test is used for comparing ranks rather than means for data that is not normally distributed. Additionally, non-normally distributed data is generally summarized using median, first and third quartiles.

3- Line 173 states, “Chi-square test was used for dichotomous variables.” Provide further clarity by indicating the comparisons which are being tested by the chi-square test.

4- Line 177 would be clarified by stating, “Sensitivity analysis was conducted for the participants with valid accelerometer measurements from all three time points, conducted with a regression analysis adjusted for baseline measures and presented as differences of the means between the groups at one- and two-years follow-up.”

5- Line 180: Again replace “calculated” with “conducted.”

6- Table 2: Clarify the statistical methods used to calculate the p-values in Table 2.

7- Line 232: Replace “numbers” with “frequency” to improve clarity.

8- Clarify if SE represents standard error or standard error of the mean.

6. PLOS authors have the option to publish the peer review history of their article (what does this mean?). If published, this will include your full peer review and any attached files.

Reviewer #1: No

Reviewer #2: No

Reviewer #3: No

---

## [Author Response · Author response to Decision Letter 0]

15 Jun 2020

Dear Associate Editor Dario Ummarino,

Thank you for the opportunity to revise our paper entitled “Physical activity in women attending a dissonance-based intervention after Roux-en-Y Gastric Bypass: A 2-year follow-up of a randomized controlled trial”. Below, please see a detailed description of our responses and ways to address each comment from the reviewers. In the manuscript, all changes performed have been highlighted with red text. In this document, all citations from the manuscript are in quotes with changes highlighted with red text.

Response: Thank you for providing this information! However, we believe that we have followed the style requirements. If we have not done so, please let us know.

2. Thank you for submitting your clinical trial to PLOS ONE and for providing the name of the registry and the registration number. The information in the registry entry suggests that your trial was registered after patient recruitment began. PLOS ONE strongly encourages authors to register all trials before recruiting the first participant in a study.

a) your reasons for your delay in registering this study (after enrolment of participants started);

b) confirmation that all related trials are registered by stating: “The authors confirm that all ongoing and related trials for this drug/intervention are registered”.

Please also ensure you report the date at which the ethics committee approved the study as well as the complete date range for patient recruitment and follow-up in the Methods section of your manuscript.

Response: Thank you for giving us the chance to clarify this! The trial was registered in February 2015 and the enrollment of participants started in January 2015. However, the enrollment that was done before the trial was registered only consisted of informing the hospitals that was participating in the trial and to start the planning of the details on how and where to recruit the participants. No analyses or outcomes have been affected or changed during or after the registration of the trial. We have now added a couple of sentences in the method section to clarify this (lines 95-101):

“The Wellbeing after Gastric Bypass (WELL-GBP) trial has been approved by the Stockholm Ethical Review Board (registration number: 2013/1847-31/2. Date at which the ethics committee approved the study: December 10th 2013). The trial was registered in February 2015 (ISRCTN16417174) and the enrollment of participants started in January 2015. However, the enrollment that was done before the trial was registered only consisted of informing the participating hospitals and to determine the details of the recruitment of participants. No analyses or outcomes have been affected or changed during or after the registration of the trial. The authors confirm that all ongoing and related trials for this drug/intervention are registered.”

Response: Thank you for the feedback! Swedish secrecy law prohibits us from making register data publicly available. The data supporting our findings were used under license and ethical approval for the current study. Readers interested in obtaining microdata or replicating our study may seek similar approvals and inquire through Statistics Sweden. For further advice see: https://www.scb.se/en/services/guidance-for-researchers-anduniversities/. However, the part where we stated “data not shown” is not a core part of the research, so we have now removed the statement “(data not shown)”.

Reviewer #1:

General comments

In this paper, authors assessed the effect of a dissonance-based intervention conducted in the first months after RYGB on habitual physical activity up to two years after surgery. They conclude that no difference was observed between physical activity at the two-year follow-up between intervention and control groups. A previous paper by the same group described the effect of the intervention at the 1-year follow-up. Strengths of the study were to use a randomized design and to assess physical activity with accelerometers. The study is well written, and the references are up to date.

My only concern is about the statistical methods used. Authors compared physical activity outcomes between the two groups at each time point (pre-surgery, one-year and two-year post-surgery) instead of comparing longitudinal changes in physical activity. By doing so, they included only participants with valid data at the 2-year follow-up and therefore excluded many participants who had valid data at baseline but not at 2 years. Mixed models might be more appropriate when several time points are available.

Response: Thank you for your comment! The reason we only measured the outcomes between the groups at each time-point, and not the longitudinal changes, is because we do not investigate the longitudinal changes in this study. We are interested to see if this intervention we have conducted had any beneficial effects on physical activity at (primarily) two years post-RYGB, which is the aim of this study, and therefore we are primarily interested in any differences between the groups at the two-year follow-up. Therefore, the longitudinal changes are of less interest for this article. We show the data for the pre- and one-year post-RYGB follow-up to show all available data for the reader, but the main aim is to look at the differences between intervention- and control groups at two years follow-up.

Specific comments

Abstract

- Line 28: Is “baseline” a pre-surgery measure? If so, “before surgery” might be more precise than “at baseline”

Response: Thank you for the comment! We have now changed all “baseline” to “pre-RYGB” in the whole manuscript, to make it clearer.

Introduction

- Line 58-67: the references are up to date but the message of this paragraph is not clear to me. The expression “In addition” (line 65) does not seem appropriate to me, “In contrast” might be more appropriate here.

What I understand from this paragraph is that: 1) exercise training has been shown effective after bariatric surgery, 2) mixed results are found regarding the spontaneous change in physical activity after bariatric surgery, 3) exercise training leads to a small and transient increase in habitual physical activity, 3) physical activity counselling might be effective to increase physical activity.

I would suggest to make a clearer distinction between changes in physical activity related to bariatric surgery and those related to exercise training after bariatric surgery.

Response: Thank you for the comment! We have now included a sentence to distinguish between physical activity (spontaneous) after surgery and interventions aiming to increase physical activity after surgery (lines 62-63): “Exercise interventions could have beneficial effect on physical activity post-surgery [14, 15]; A randomized controlled trial (RCT), investigating if exercise after surgery could increase health…”.

We then changed “in addition” to “also” on line 66 (previous line 65), because of the change we made to the paragraph shown above: “Also, individualized physical activity counseling may increase physical activity post-bariatric surgery [15].” 

- Line 91: “objectively-measured” rather than “objective measured”

Response: Thank you for letting us improve the grammar! We have now changed it according to your suggestions (line 91): “The aim of this study was to examine if there were any differences in objectively-measured physical activity levels…”.

Methods

- Line 98-99: “we analysed how the intervention affected physical activity”

Response: Thank you for letting us improve the grammar! We have now changed it according to your suggestions (line 104): “In this paper we analyzed how the intervention affected physical activity…”.

Results

- Tables do not fit in the pdf. I cannot comment the tables.

Response: Thank you for acknowledging this! We assumed it would be able to see the tables anyway. But now we have changed the set of the pages in the document to “landscape” where all the tables are, in order for you to be able to see them. We hope that will work.

- The authors compared physical activity outcomes between intervention and control group at all times (Table 2). This method results in the exclusion of many participants who do no hate valid measures at the 2-year follow-up. It would have been more appropriate to compare changes in physical activity between groups. Mixed models are usually used in this situation.

Response: Thank you for the comment! However, we do not show data for the participants who had valid measurements at all time points. As stated in the first row of the table, number of participants with valid measurements are seen (n=77 for example) for each time-point. As you can see, there are more participants who have valid data at the two-year follow-up than at pre-RYGB. This is because not all participants had time to wear an accelerometer at pre-RYGB, and we have stated that in the text under the table. The aim was to analyze differences between groups at two-years follow-up, which is why we included only participants who had valid measurements at the two-years follow-up. Thus, table 2 shows participants who had valid measurements at the two-year follow-up, and if those participants also had valid measurements at pre- and one-year post-RYGB, they were included in the pre- and one-year analyses as well. This was so we would not exclude participants who had valid measurements at two-years, but not at the pre- or one-year follow-up. This is, as mentioned, described in the text under table 2, as well as in the method section (statistical analysis) on lines 179-182.

Discussion

- Lines 307-309: Are the results similar in other contexts (eg in the studies cited in the introduction)?

Response: Thank you for the comment! However, we are not sure what you mean. There have not been many previous studies investigating dissonance-based interventions on physical activity (the ones we know of is what we have cited and includes female college students) and it has never, to our best knowledge, been conducted on bariatric surgery patients. We have stated this in the manuscript (on lines 37-38, 79-80, 343-345 and 411-412) and therefore we cannot conclude whether or not the results are similar in other contexts.

Reviewer #2:

Thank you for the opportunity to review the manuscript entitled ‘Physical activity in women attending a dissonance-based intervention after Roux-en-Y- Gastric Bypass: A 2-year follow-up of a randomized controlled trial’. This is an interesting paper looking at objectively measured physical activity in a sample of women whom have undergone RYGB, this is a large sample given the population and topic area (physical activity) that the authors are researching in. It is also an area of research which is in its infancy and a useful addition to the literature.

Please see my comments in the document attached.

General comments

Overall a good well written manuscript, I have the following questions I would like you to consider;

1. You have undertaken a dissonance-based intervention comprising four sessions that focus on different topic areas. One of these focus on physical activity, nowhere in this manuscript can I see how many of the individuals analysed attended this physical activity session. Surely it is important to know how many attended the physical activity session; you wouldn’t typically expect physical activity to increase when compared to normal care if they have not been given an additional physical activity intervention. Did you look at data from those who attended the physical activity session? 

Response: Thank you for this excellent comment! Of course, you are right, it is of importance to look at how many of the patients attended the physical activity session. We have now done such an analysis, and 61.6% (n=61) of the participants in the intervention group attended the session about physical activity. However, it did not change any of the results. We did per protocol analysis and included a few sentences about this in the manuscript:

In the section Methods – Statistical analysis (lines 196-198): “Per protocol analysis was also performed to compare the participants in the intervention group who had participated in the session about physical activity, to the participants who had not participated in that session.”

In the section Results – descriptive statistics (lines 215-216): “Of the participants in the intervention group, 61.6% (n=61) had attended the session where physical activity had been discussed.”

In the section Results – Per protocol analysis (lines 269-273): “Per protocol analysis of the participants attending the session about physical activity (n=61) versus participants in the intervention group who did not attend that session (n=38), showed no significant differences of clinical importance in any of the physical activity outcomes that changed any of the results. There were also no differences in any of the pre-RYGB characteristics between these two groups.” 

2. All tables were cut off so I was unable to read half of the table.

Response: Thank you for letting us know! We have now adjusted this, so the pages where the tables are, are now set into “landscape” in order for the whole tables to be seen.

Specific comments

Page 1 line 58 – instead of ‘inconclusive’ should this be ‘limited’

Response: Thank you for the comment! We have now changed the sentence according to your suggestion (line 58): “However, data remains limited on this issue.”

Page 2 line 74 – remove for example

Response: Thank you for this comment! We have now removed the words and the sentence is now (Lines 73-75): “Dissonance-based interventions have shown effects in behavioral change; for healthy physical activity behaviors [19], smoking cessation….”

Page 2 line 77 – instead of ‘Also’ should this be ‘However’

Response: Thank you for the comment! We have now changed the sentence according to your suggestion (line 77): “However, the effect of an already existing program….”

Page 2 line 81 – change ‘surgeries’ to ‘surgery procedures’

Response: Thank you for the comment! We have now changed the sentence according to your suggestion (line 81): “In Sweden, hospitals performing bariatric surgery procedures have standard…”

Page 2 lines 87-90 – instead of ‘with the aim to’ potentially change to ‘aiming to’, this is a long sentence.

Response: Thank you for the comment! We have now changed the sentence according to your suggestion (line 89): “…a dissonance-based group intervention program targeting RYGB-treated women, aiming to prevent a decline in…”

Page 5 line 98-99 – this sentence needs to be slightly reworded or add how after ‘analyse’

Response: Thank you for the comment! We have now changed the sentence according to your suggestion (line 104): “In this paper we analyzed how the intervention affected….”

Page 5 line 109 – Was wearing the accelerometer optional? ‘ were offered to wear’ makes it sound optional

Response: Thank you for the comment! It was optional to wear the accelerometer, as the questionnaires measuring HRQoL (among others) were the main thing for this intervention (questionnaires are not included in this study. We have now added a sentence about this in the section Method – Recruitment and participants (lines 117-118): “To wear the accelerometer was optional for the participants.”

Page 6 line 117 Re-wording suggestion ‘ participants who had provided informed consent and baseline data…’

Response: Thank you for letting us improve the wording! We have now changed the sentence according to your suggestion, as well as clarified the pre-RYGB data (lines 126-127): 

“Approximately two months post-RYGB, participants who had provided informed consent and pre-RYGB data (questionnaires about HRQoL, among others (not included in this study)) were…”

Page 6 line 127 Trained facilitators – include what the training comprised of and in what ‘teaching’ style was the programme delivered?

Response: Thank you for the opportunity to clarify this! We have now added a paragraph in that sentence to describe what the training contained (line 138): 

“…led by a facilitator who had been trained in dissonance-based theory (individual readings on the topic together with informational meetings with a psychologist) and followed an intervention manual.”

Page 6 line 135-136 – the participant was considered as having received the intervention if they attended three sessions was it any three sessions? If so state this.

Response: Thank you for letting us clarify this! Yes, it was at least three of any of the four sessions. We have now clarified this in that sentence (line 147): “If a participant had attended at least three out of any of the four sessions…”

Page 6 line 139-140 – I think it would be useful to mention the topics covered in the physical activity session.

Response: Thank you for your valuable input! We have now included a sentence about what were discussed during the physical activity session in the section Methods – intervention (lines 152-155): “During the session on physical activity, discussions were focused on what sedentary behavior and physical activity is, how one can increase their daily physical activity, difficulties and barriers of physical activity after bariatric surgery and how to overcome these barriers.”

Page 6 line 139-140 - Did you refine the programme based on the pilot feedback – this may also be worth mentioning

Response: Thank you for letting us clarify this! Yes, we did some minor changes after the feed-back from the pilot study. We have now included a sentence about this in the section Method – Intervention (line 151): “Minor changes were done to the manuscript after the feed-back.”

Page 7 line 144 – was the accelerometer worn on the dominant or non-dominant wrist? This should be reported.

Response: Thank you for this input in clarifying the manuscript! The accelerometer was worn on the right hip, and we have now added this information to the sentence you are referring to, in section Method – Outcomes (line 158): “…were asked to wear the ActiGraph GT3X+ accelerometer on the right hip during all waking hours…”

Page 7 line 146 – why the 10 hour threshold? Is there a reference for this?

Response: When having the criteria for using the accelerometer during all waking hours, a threshold of at least 10 hours is common, as it is a majority of the hours of the day spent awake. There is already a reference on this in the section Discussion – Strengths and limitations (lines 367-369): “Inclusion criterion for valid wear time was set to at least three days with a minimum of 10-hour wear time per day, as three to four wear time days has shown to be sufficient for achieving 80% reliability of MVPA [43].”

Page 9 table half missing and the variables in the table are not clear if they are baseline or 2 years post.

Response: Thank you for the comment! The pages where the tables are, are now set into “landscape”, in order for the whole tables to be seen. As stated in the heading of the table 1, it is pre-RYGB measures that are stated in the table. To clarify, we have now added more information in the table to make it clearer that it is pre-RYGB measures (if not stated otherwise) in Table 1 (page 11, lines 231-232).

Page 10 line 225 – what do you mean by non bouts? 1 minute bouts?

Response: Thank you for letting us clarify this! In the manuscript we use both 10-min bouts and non-bouts, where non-bouts mean just like it sounds: the MVPA have been measured as it is and do not have any requirement that the MVPA should have been going on for any set amount of time to be counted as MVPA, where 10-min bouts mean the participants had to be active for at least 10-min in MVPA-level to be counted in that outcome. We therefore write “non-bouts” in the manuscript to distinguish which outcome we are referring to, in order to not make any confusing. Therefore, non-bouts mean that there are no bouts whatsoever in that outcome, but that all time spent in MVPA have been accounted for.

Page 10-11 Table 2 format

Response: Thank you for letting us know about the tables! As mentioned above, we have now set the pages where the tables are into “landscape”, hopefully you will be able to see the tables now.

Page 11 line 232 – the sentence starts with presented – what is this sentence referring to?

Response: Thank you for letting us clarify this section! The sentence belongs to the description of Table 2 (the text under the table). To make it clearer, we have now moved up that sentence to the line above, so it is clearer that it is a part of the Table 2 (line 253).

Page 11 line 244 – a sedentary time difference between groups of 64 minutes per day seems large, you have stated that it is not relevant from a clinical perspective… if this is correct you need to reference this. This is implying that the individuals are more sedentary as a result of attending the programme, is this correct?

Response: Thank you for the comment! What we meant was that this difference between the groups was also significant at pre-RYGB, and therefore may not be relevant or something to put too much emphasis on. We also already have stated in the section Discussion – Limitations (lines 392-394): “Second, smaller sample sizes were used when conducting per protocol and sensitivity analyses. Consequently, the power might be too small to detect between-group differences.”

But, to clarify this in the manuscript, we have added to section Results – Per protocol analysis that the significant difference was there also at pre-RYGB, and we added that it may not be relevant from a clinical perspective (lines 264-268): “Those who received the intervention were more sedentary (522.5 min/day, SE=17.2) than the control group (458.8 min/day, SE=12.3) (p=0.002, d=-0.58) at two-years post-RYGB (Table 2 in S2 Appendix). However, this difference was also significantly different at pre-RYGB (p=0.039), and therefore this difference may not be relevant from a clinical perspective.”

Page 13 discussion – I think the difference in sedentary time (increase in intervention group) should be mentioned in the results and justified.

Response: Thank you for the comment! Do you refer to the section of main results (lines 309-310)? In that case, unfortunately, we do not agree with you. The main results are the Intention-to-treat analysis, Table 2 (as those analysis have enough power, which the per protocol analysis do not have due to small sample sizes, which, as already mentioned above, are stated in the Discussion-section (lines 392-394)), and this study focuses on the differences between the groups at the two-years follow-up (as we want to see if our intervention had have any effect), and not the longitudinal differences from the time-points. There is no statistically significant difference between the groups in sedentary time at the two-year follow-up in the ITT-analysis in Table 2. And if you are referring to the difference in the intervention group from pre-RYGB to two-years post-RYGB in Table 2 (the ITT-analysis), it is only a difference if 16 min.

Page 13 line 286 – when referring to the other research, it would be useful to know when these studies were undertaken post-surgery was it 2 years or 6 months etc?

Response: Thank you for your valuable input! We have now included in that sentence the time of the start and the length of the interventions right after the sentence you are referring to in section Discussion – Previous research (lines 315-317): “The interventions included in that review started pre-surgery or between three to 102 months post-surgery and the lengths of the interventions were between six and 12 weeks long [15].”

Page 13 line 294 – physical activity increased by how much?

Response: Thank you for this input! We have now added that information in a paragraph right after the end of that sentence you are referring to (lines 326-327): “…(counselling and pedometers = 6 787 steps/day, pedometers only = 5 325 steps/day and standard care = 5253 steps/day) [34].”

Page 14 line 314 – attendance at at least one session, was this one session the physical activity session or not? If not why not? If not how many of those included attended that session?

If they did not attend the physical activity session this is likely why you didn’t see a change and previous research in this population reports exercise education alone is insufficient for increasing physical activity.

Response: Thank you for this comment! This comment is in line with a previous comment from you above, about how many attended the session about physical activity. When “at least one session” is mentioned in the manuscript, it is at least one of any of the four sessions in the intervention. We have added, after input from you in the comment above, information and per protocol analysis on how many participants attended the session about physical activity and the results from the per protocol analysis. We therefore refer to our answer on that question above where we have stated all the changes we made to the manuscript according to your comment. We have now, after this comment, also clarified that we mean at least any of the four sessions (line 347): “… 73% attended at least one session (any of the four sessions)…”

The same addition (any of the four sessions) have been added in the section Results – Descriptive analysis (line 225): “…had attended at least one group session (any of the four sessions),…”.

Limitation – may not have attended the physical activity session

Response: Thank you for this valuable input! You are right, we have not mentioned that as a limitation. We have now added that to the section Discussion – Limitations (lines 402-404): “Fifth, only 61.6% of the participants in the intervention group attended the session about physical activity, which might have had impact on the results.”

You have mentioned you were likely not powered, have you looked at your data to see how many individuals you would have needed to be powered?

Response: Thank you for this comment! However, no we have not looked at that data, as we calculated the power of the intervention according to HRQoL (which is part of another yet not published article) as the initial aim of this intervention was to try to maintain/prevent the decline in HRQoL post-surgery. 

Fig 1 – include amount of individuals who attended the physical activity session

Response: Thank you for this input! We have now included this information in the figure caption of Figure 1 (lines 120-122): “Of the included participants in the two-year follow-up belonging to the intervention group, 61 of them (61.6%) had attended the session about physical activity.”

Reviewer #3:

Two-year follow-up results on patients activity levels were summarized from a two-arm randomized controlled clinical trial which aimed to increase quality of life following Roux-en-Y gastric bypass surgery through dissonance-based group intervention. At two-years follow-up, no significant differences in physical activity levels were observed between the intervention and control arms.

Minor revisions:

1- Cohen’s d is valid when the distribution of the data is normal. Indicate if the normality assumption was met.

2- Line 172: Technically the Kruskal-Wallis H test is used for comparing ranks rather than means for data that is not normally distributed. Additionally, non-normally distributed data is generally summarized using median, first and third quartiles.

Response: Thank you for these comments! As these two comments are related, we will answer them together. We have now added a paragraph in the section Method – Statistical analysis on information about normality assumptions and why we chose to use Kruskal-Wallis H test and Cohen’s D (lines 185-189): “Normal distribution was evaluated graphically and tested with Shapiro-Wilk test. The variables were approximately normally distributed, however not perfectly. We therefore present the results as means and use Cohen’s d to quantify intervention effect sizes [31], but because of the lack of complete normality, we also used Kruskal-Wallis H test to test for differences in medians between the groups.”

We also added information under Table 2 (lines 254-256): “…p-value for the difference in medians between the two groups (Kruskal-Wallis H test) at pre-RYGB, one- and two-years post-RYGB surgery. Effect sizes at two years measured with Cohen’s d (95 % CI). *Normal distribution assumption rejected according to Shapiro-Wilk test (p<0.05).”

3- Line 173 states, “Chi-square test was used for dichotomous variables.” Provide further clarity by indicating the comparisons which are being tested by the chi-square test.

Response: Thank you for the opportunity to clarify this! Variables that were dichotomous were for example “smokers or not”, “university level or not” and “meeting physical activity recommendations or not”. We have now included thin in the section Method – Statistical analysis (lines 191-192): “Chi-square test was used for dichotomous variables, for example if a participant smoked or not, had university level of education or not, and meeting the physical activity recommendations or not.”

4- Line 177 would be clarified by stating, “Sensitivity analysis was conducted for the participants with valid accelerometer measurements from all three time points, conducted with a regression analysis adjusted for baseline measures and presented as differences of the means between the groups at one- and two-years follow-up.”

Response: Thank you for letting us clarify this sentence! We have now changed it according to your suggestion (line 198-201): “Sensitivity analysis was conducted for the participants with valid accelerometer measurements from all three time points, conducted with a regression analysis adjusted for pre-RYGB measures and presented as differences of the means between the groups at one- and two-years follow-up.”

5- Line 180: Again replace “calculated” with “conducted.”

Response: Thank you for letting us improve the sentence! We have now changed it according to your suggestion (line 202): “Additionally, a sensitivity analysis was conducted for all participants who had ≥5 valid days…”

6- Table 2: Clarify the statistical methods used to calculate the p-values in Table 2.

Response: Thank you for the comment! We have now clarified this under Table 2 and added how we calculated the p-value (line 254): “p-value for the difference in medians between the two groups (Kruskal-Wallis H test) at pre-RYGB, one- and two-years post-RYGB surgery.”

We have also added the same information under S2 Appendix and S3 Appendix.

7- Line 232: Replace “numbers” with “frequency” to improve clarity.

Response: Thank you for the comment! We have now changed the sentence according to your suggestion in the text under Table 2 (line 254): “…or frequency (percent) for each subscale…”

We have also added that same information under S2 Appendix and S3 Appendix.

8- Clarify if SE represents standard error or standard error of the mean.

Response: Thank you for the comment! We do mean Standard Error of the mean.

---

## [Decision Letter · Decision Letter 1]

17 Aug 2020

PONE-D-20-01663R1

Physical activity in women attending a dissonance-based intervention after Roux-en-Y Gastric Bypass: A 2-year follow-up of a randomized controlled trial

PLOS ONE

Dear Dr. Berglind,

Thank you for submitting your manuscript to PLOS ONE. After careful consideration, we feel that it has merit but does not fully meet PLOS ONE’s publication criteria as it currently stands. Therefore, we invite you to submit a revised version of the manuscript that addresses the points raised during the review process.

As you can see from the comments included at the end of this letter, I am pleased to say that the reviewers are satisfied that the previously raised concerns have been adequately addressed. There is just one minor point from reviewer 3 regarding the statistical test used.

In addition, in order to fully comply with our editorial requirements for clinical trial submissions, please upload the full clinical trial protocol in the original language that was submitted to your local ethics committee. 

A rebuttal letter that responds to each point raised in this letter. You should upload this letter as a separate file labeled 'Response to Reviewers'.A marked-up copy of your manuscript that highlights changes made to the original version. You should upload this as a separate file labeled 'Revised Manuscript with Track Changes'.An unmarked version of your revised paper without tracked changes. You should upload this as a separate file labeled 'Manuscript'.

We look forward to receiving your revised manuscript.

Kind regards,

Dario Ummarino, Ph.D.

Associate Editor

PLOS ONE

Reviewers' comments:

Reviewer's Responses to Questions

**Comments to the Author**

1. If the authors have adequately addressed your comments raised in a previous round of review and you feel that this manuscript is now acceptable for publication, you may indicate that here to bypass the “Comments to the Author” section, enter your conflict of interest statement in the “Confidential to Editor” section, and submit your "Accept" recommendation.

Reviewer #1: (No Response)

Reviewer #3: (No Response)

Reviewer #4: All comments have been addressed

2. Is the manuscript technically sound, and do the data support the conclusions?

Reviewer #1: Yes

Reviewer #3: Yes

Reviewer #4: Partly

3. Has the statistical analysis been performed appropriately and rigorously? 

Reviewer #1: Yes

Reviewer #3: Yes

Reviewer #4: Yes

4. Have the authors made all data underlying the findings in their manuscript fully available?

Reviewer #1: Yes

Reviewer #3: No

Reviewer #4: Yes

5. Is the manuscript presented in an intelligible fashion and written in standard English?

Reviewer #1: Yes

Reviewer #3: Yes

Reviewer #4: Yes

6. Review Comments to the Author

Reviewer #1: (No Response)

Reviewer #3: Technically, the Kruskal-Wallis test compares ranks rather than medians.

Reviewer #4: The authors should be commended for their thorough responses to detailed critiques, in particular the explicit stating of the study limitations.

7. PLOS authors have the option to publish the peer review history of their article (what does this mean?). If published, this will include your full peer review and any attached files.

Reviewer #1: No

Reviewer #3: No

Reviewer #4: No

---

## [Author Response · Author response to Decision Letter 1]

10 Sep 2020

PLOS ONE

Manuscript number PONE-D-20-01663

Dear Associate Editor Dario Ummarino,

Thank you for the opportunity to revise our paper entitled “Physical activity in women attending a dissonance-based intervention after Roux-en-Y Gastric Bypass: A 2-year follow-up of a randomized controlled trial”. Below, please see a detailed description of our responses and ways to address each comment from the reviewers. In the manuscript, all changes performed have been highlighted with red text. In this document, all citations from the manuscript are in quotes with changes highlighted with red text.

Response: Thank you for providing this information! However, we believe that we have followed the style requirements. If we have not done so, please let us know.

2. Thank you for submitting your clinical trial to PLOS ONE and for providing the name of the registry and the registration number. The information in the registry entry suggests that your trial was registered after patient recruitment began. PLOS ONE strongly encourages authors to register all trials before recruiting the first participant in a study.

a) your reasons for your delay in registering this study (after enrolment of participants started);

b) confirmation that all related trials are registered by stating: “The authors confirm that all ongoing and related trials for this drug/intervention are registered”.

Please also ensure you report the date at which the ethics committee approved the study as well as the complete date range for patient recruitment and follow-up in the Methods section of your manuscript.

Response: Thank you for giving us the chance to clarify this! The trial was registered in February 2015 and the enrollment of participants started in January 2015. However, the enrollment that was done before the trial was registered only consisted of informing the hospitals that was participating in the trial and to start the planning of the details on how and where to recruit the participants. No analyses or outcomes have been affected or changed during or after the registration of the trial. We have now added a couple of sentences in the method section to clarify this (lines 95-101):

“The Wellbeing after Gastric Bypass (WELL-GBP) trial has been approved by the Stockholm Ethical Review Board (registration number: 2013/1847-31/2. Date at which the ethics committee approved the study: December 10th 2013). The trial was registered in February 2015 (ISRCTN16417174) and the enrollment of participants started in January 2015. However, the enrollment that was done before the trial was registered only consisted of informing the participating hospitals and to determine the details of the recruitment of participants. No analyses or outcomes have been affected or changed during or after the registration of the trial. The authors confirm that all ongoing and related trials for this drug/intervention are registered.”

Response: Thank you for the feedback! Swedish secrecy law prohibits us from making register data publicly available. The data supporting our findings were used under license and ethical approval for the current study. Readers interested in obtaining microdata or replicating our study may seek similar approvals and inquire through Statistics Sweden. For further advice see: https://www.scb.se/en/services/guidance-for-researchers-anduniversities/. However, the part where we stated “data not shown” is not a core part of the research, so we have now removed the statement “(data not shown)”.

Reviewer #1:

General comments

In this paper, authors assessed the effect of a dissonance-based intervention conducted in the first months after RYGB on habitual physical activity up to two years after surgery. They conclude that no difference was observed between physical activity at the two-year follow-up between intervention and control groups. A previous paper by the same group described the effect of the intervention at the 1-year follow-up. Strengths of the study were to use a randomized design and to assess physical activity with accelerometers. The study is well written, and the references are up to date.

My only concern is about the statistical methods used. Authors compared physical activity outcomes between the two groups at each time point (pre-surgery, one-year and two-year post-surgery) instead of comparing longitudinal changes in physical activity. By doing so, they included only participants with valid data at the 2-year follow-up and therefore excluded many participants who had valid data at baseline but not at 2 years. Mixed models might be more appropriate when several time points are available.

Response: Thank you for your comment! The reason we only measured the outcomes between the groups at each time-point, and not the longitudinal changes, is because we do not investigate the longitudinal changes in this study. We are interested to see if this intervention we have conducted had any beneficial effects on physical activity at (primarily) two years post-RYGB, which is the aim of this study, and therefore we are primarily interested in any differences between the groups at the two-year follow-up. Therefore, the longitudinal changes are of less interest for this article. We show the data for the pre- and one-year post-RYGB follow-up to show all available data for the reader, but the main aim is to look at the differences between intervention- and control groups at two years follow-up.

Specific comments

Abstract

- Line 28: Is “baseline” a pre-surgery measure? If so, “before surgery” might be more precise than “at baseline”

Response: Thank you for the comment! We have now changed all “baseline” to “pre-RYGB” in the whole manuscript, to make it clearer.

Introduction

- Line 58-67: the references are up to date but the message of this paragraph is not clear to me. The expression “In addition” (line 65) does not seem appropriate to me, “In contrast” might be more appropriate here.

What I understand from this paragraph is that: 1) exercise training has been shown effective after bariatric surgery, 2) mixed results are found regarding the spontaneous change in physical activity after bariatric surgery, 3) exercise training leads to a small and transient increase in habitual physical activity, 3) physical activity counselling might be effective to increase physical activity.

I would suggest to make a clearer distinction between changes in physical activity related to bariatric surgery and those related to exercise training after bariatric surgery.

Response: Thank you for the comment! We have now included a sentence to distinguish between physical activity (spontaneous) after surgery and interventions aiming to increase physical activity after surgery (lines 62-63): “Exercise interventions could have beneficial effect on physical activity post-surgery [14, 15]; A randomized controlled trial (RCT), investigating if exercise after surgery could increase health…”.

We then changed “in addition” to “also” on line 66 (previous line 65), because of the change we made to the paragraph shown above: “Also, individualized physical activity counseling may increase physical activity post-bariatric surgery [15].” 

- Line 91: “objectively-measured” rather than “objective measured”

Response: Thank you for letting us improve the grammar! We have now changed it according to your suggestions (line 91): “The aim of this study was to examine if there were any differences in objectively-measured physical activity levels…”.

Methods

- Line 98-99: “we analysed how the intervention affected physical activity”

Response: Thank you for letting us improve the grammar! We have now changed it according to your suggestions (line 104): “In this paper we analyzed how the intervention affected physical activity…”.

Results

- Tables do not fit in the pdf. I cannot comment the tables.

Response: Thank you for acknowledging this! We assumed it would be able to see the tables anyway. But now we have changed the set of the pages in the document to “landscape” where all the tables are, in order for you to be able to see them. We hope that will work.

- The authors compared physical activity outcomes between intervention and control group at all times (Table 2). This method results in the exclusion of many participants who do no hate valid measures at the 2-year follow-up. It would have been more appropriate to compare changes in physical activity between groups. Mixed models are usually used in this situation.

Response: Thank you for the comment! However, we do not show data for the participants who had valid measurements at all time points. As stated in the first row of the table, number of participants with valid measurements are seen (n=77 for example) for each time-point. As you can see, there are more participants who have valid data at the two-year follow-up than at pre-RYGB. This is because not all participants had time to wear an accelerometer at pre-RYGB, and we have stated that in the text under the table. The aim was to analyze differences between groups at two-years follow-up, which is why we included only participants who had valid measurements at the two-years follow-up. Thus, table 2 shows participants who had valid measurements at the two-year follow-up, and if those participants also had valid measurements at pre- and one-year post-RYGB, they were included in the pre- and one-year analyses as well. This was so we would not exclude participants who had valid measurements at two-years, but not at the pre- or one-year follow-up. This is, as mentioned, described in the text under table 2, as well as in the method section (statistical analysis) on lines 179-182.

Discussion

- Lines 307-309: Are the results similar in other contexts (eg in the studies cited in the introduction)?

Response: Thank you for the comment! However, we are not sure what you mean. There have not been many previous studies investigating dissonance-based interventions on physical activity (the ones we know of is what we have cited and includes female college students) and it has never, to our best knowledge, been conducted on bariatric surgery patients. We have stated this in the manuscript (on lines 37-38, 79-80, 343-345 and 411-412) and therefore we cannot conclude whether or not the results are similar in other contexts.

Reviewer #2:

Thank you for the opportunity to review the manuscript entitled ‘Physical activity in women attending a dissonance-based intervention after Roux-en-Y- Gastric Bypass: A 2-year follow-up of a randomized controlled trial’. This is an interesting paper looking at objectively measured physical activity in a sample of women whom have undergone RYGB, this is a large sample given the population and topic area (physical activity) that the authors are researching in. It is also an area of research which is in its infancy and a useful addition to the literature.

Please see my comments in the document attached.

General comments

Overall a good well written manuscript, I have the following questions I would like you to consider;

1. You have undertaken a dissonance-based intervention comprising four sessions that focus on different topic areas. One of these focus on physical activity, nowhere in this manuscript can I see how many of the individuals analysed attended this physical activity session. Surely it is important to know how many attended the physical activity session; you wouldn’t typically expect physical activity to increase when compared to normal care if they have not been given an additional physical activity intervention. Did you look at data from those who attended the physical activity session? 

Response: Thank you for this excellent comment! Of course, you are right, it is of importance to look at how many of the patients attended the physical activity session. We have now done such an analysis, and 61.6% (n=61) of the participants in the intervention group attended the session about physical activity. However, it did not change any of the results. We did per protocol analysis and included a few sentences about this in the manuscript:

In the section Methods – Statistical analysis (lines 196-198): “Per protocol analysis was also performed to compare the participants in the intervention group who had participated in the session about physical activity, to the participants who had not participated in that session.”

In the section Results – descriptive statistics (lines 215-216): “Of the participants in the intervention group, 61.6% (n=61) had attended the session where physical activity had been discussed.”

In the section Results – Per protocol analysis (lines 269-273): “Per protocol analysis of the participants attending the session about physical activity (n=61) versus participants in the intervention group who did not attend that session (n=38), showed no significant differences of clinical importance in any of the physical activity outcomes that changed any of the results. There were also no differences in any of the pre-RYGB characteristics between these two groups.” 

2. All tables were cut off so I was unable to read half of the table.

Response: Thank you for letting us know! We have now adjusted this, so the pages where the tables are, are now set into “landscape” in order for the whole tables to be seen.

Specific comments

Page 1 line 58 – instead of ‘inconclusive’ should this be ‘limited’

Response: Thank you for the comment! We have now changed the sentence according to your suggestion (line 58): “However, data remains limited on this issue.”

Page 2 line 74 – remove for example

Response: Thank you for this comment! We have now removed the words and the sentence is now (Lines 73-75): “Dissonance-based interventions have shown effects in behavioral change; for healthy physical activity behaviors [19], smoking cessation….”

Page 2 line 77 – instead of ‘Also’ should this be ‘However’

Response: Thank you for the comment! We have now changed the sentence according to your suggestion (line 77): “However, the effect of an already existing program….”

Page 2 line 81 – change ‘surgeries’ to ‘surgery procedures’

Response: Thank you for the comment! We have now changed the sentence according to your suggestion (line 81): “In Sweden, hospitals performing bariatric surgery procedures have standard…”

Page 2 lines 87-90 – instead of ‘with the aim to’ potentially change to ‘aiming to’, this is a long sentence.

Response: Thank you for the comment! We have now changed the sentence according to your suggestion (line 89): “…a dissonance-based group intervention program targeting RYGB-treated women, aiming to prevent a decline in…”

Page 5 line 98-99 – this sentence needs to be slightly reworded or add how after ‘analyse’

Response: Thank you for the comment! We have now changed the sentence according to your suggestion (line 104): “In this paper we analyzed how the intervention affected….”

Page 5 line 109 – Was wearing the accelerometer optional? ‘ were offered to wear’ makes it sound optional

Response: Thank you for the comment! It was optional to wear the accelerometer, as the questionnaires measuring HRQoL (among others) were the main thing for this intervention (questionnaires are not included in this study. We have now added a sentence about this in the section Method – Recruitment and participants (lines 117-118): “To wear the accelerometer was optional for the participants.”

Page 6 line 117 Re-wording suggestion ‘ participants who had provided informed consent and baseline data…’

Response: Thank you for letting us improve the wording! We have now changed the sentence according to your suggestion, as well as clarified the pre-RYGB data (lines 126-127): 

“Approximately two months post-RYGB, participants who had provided informed consent and pre-RYGB data (questionnaires about HRQoL, among others (not included in this study)) were…”

Page 6 line 127 Trained facilitators – include what the training comprised of and in what ‘teaching’ style was the programme delivered?

Response: Thank you for the opportunity to clarify this! We have now added a paragraph in that sentence to describe what the training contained (line 138): 

“…led by a facilitator who had been trained in dissonance-based theory (individual readings on the topic together with informational meetings with a psychologist) and followed an intervention manual.”

Page 6 line 135-136 – the participant was considered as having received the intervention if they attended three sessions was it any three sessions? If so state this.

Response: Thank you for letting us clarify this! Yes, it was at least three of any of the four sessions. We have now clarified this in that sentence (line 147): “If a participant had attended at least three out of any of the four sessions…”

Page 6 line 139-140 – I think it would be useful to mention the topics covered in the physical activity session.

Response: Thank you for your valuable input! We have now included a sentence about what were discussed during the physical activity session in the section Methods – intervention (lines 152-155): “During the session on physical activity, discussions were focused on what sedentary behavior and physical activity is, how one can increase their daily physical activity, difficulties and barriers of physical activity after bariatric surgery and how to overcome these barriers.”

Page 6 line 139-140 - Did you refine the programme based on the pilot feedback – this may also be worth mentioning

Response: Thank you for letting us clarify this! Yes, we did some minor changes after the feed-back from the pilot study. We have now included a sentence about this in the section Method – Intervention (line 151): “Minor changes were done to the manuscript after the feed-back.”

Page 7 line 144 – was the accelerometer worn on the dominant or non-dominant wrist? This should be reported.

Response: Thank you for this input in clarifying the manuscript! The accelerometer was worn on the right hip, and we have now added this information to the sentence you are referring to, in section Method – Outcomes (line 158): “…were asked to wear the ActiGraph GT3X+ accelerometer on the right hip during all waking hours…”

Page 7 line 146 – why the 10 hour threshold? Is there a reference for this?

Response: When having the criteria for using the accelerometer during all waking hours, a threshold of at least 10 hours is common, as it is a majority of the hours of the day spent awake. There is already a reference on this in the section Discussion – Strengths and limitations (lines 367-369): “Inclusion criterion for valid wear time was set to at least three days with a minimum of 10-hour wear time per day, as three to four wear time days has shown to be sufficient for achieving 80% reliability of MVPA [43].”

Page 9 table half missing and the variables in the table are not clear if they are baseline or 2 years post.

Response: Thank you for the comment! The pages where the tables are, are now set into “landscape”, in order for the whole tables to be seen. As stated in the heading of the table 1, it is pre-RYGB measures that are stated in the table. To clarify, we have now added more information in the table to make it clearer that it is pre-RYGB measures (if not stated otherwise) in Table 1 (page 11, lines 231-232).

Page 10 line 225 – what do you mean by non bouts? 1 minute bouts?

Response: Thank you for letting us clarify this! In the manuscript we use both 10-min bouts and non-bouts, where non-bouts mean just like it sounds: the MVPA have been measured as it is and do not have any requirement that the MVPA should have been going on for any set amount of time to be counted as MVPA, where 10-min bouts mean the participants had to be active for at least 10-min in MVPA-level to be counted in that outcome. We therefore write “non-bouts” in the manuscript to distinguish which outcome we are referring to, in order to not make any confusing. Therefore, non-bouts mean that there are no bouts whatsoever in that outcome, but that all time spent in MVPA have been accounted for.

Page 10-11 Table 2 format

Response: Thank you for letting us know about the tables! As mentioned above, we have now set the pages where the tables are into “landscape”, hopefully you will be able to see the tables now.

Page 11 line 232 – the sentence starts with presented – what is this sentence referring to?

Response: Thank you for letting us clarify this section! The sentence belongs to the description of Table 2 (the text under the table). To make it clearer, we have now moved up that sentence to the line above, so it is clearer that it is a part of the Table 2 (line 253).

Page 11 line 244 – a sedentary time difference between groups of 64 minutes per day seems large, you have stated that it is not relevant from a clinical perspective… if this is correct you need to reference this. This is implying that the individuals are more sedentary as a result of attending the programme, is this correct?

Response: Thank you for the comment! What we meant was that this difference between the groups was also significant at pre-RYGB, and therefore may not be relevant or something to put too much emphasis on. We also already have stated in the section Discussion – Limitations (lines 392-394): “Second, smaller sample sizes were used when conducting per protocol and sensitivity analyses. Consequently, the power might be too small to detect between-group differences.”

But, to clarify this in the manuscript, we have added to section Results – Per protocol analysis that the significant difference was there also at pre-RYGB, and we added that it may not be relevant from a clinical perspective (lines 264-268): “Those who received the intervention were more sedentary (522.5 min/day, SE=17.2) than the control group (458.8 min/day, SE=12.3) (p=0.002, d=-0.58) at two-years post-RYGB (Table 2 in S2 Appendix). However, this difference was also significantly different at pre-RYGB (p=0.039), and therefore this difference may not be relevant from a clinical perspective.”

Page 13 discussion – I think the difference in sedentary time (increase in intervention group) should be mentioned in the results and justified.

Response: Thank you for the comment! Do you refer to the section of main results (lines 309-310)? In that case, unfortunately, we do not agree with you. The main results are the Intention-to-treat analysis, Table 2 (as those analysis have enough power, which the per protocol analysis do not have due to small sample sizes, which, as already mentioned above, are stated in the Discussion-section (lines 392-394)), and this study focuses on the differences between the groups at the two-years follow-up (as we want to see if our intervention had have any effect), and not the longitudinal differences from the time-points. There is no statistically significant difference between the groups in sedentary time at the two-year follow-up in the ITT-analysis in Table 2. And if you are referring to the difference in the intervention group from pre-RYGB to two-years post-RYGB in Table 2 (the ITT-analysis), it is only a difference if 16 min.

Page 13 line 286 – when referring to the other research, it would be useful to know when these studies were undertaken post-surgery was it 2 years or 6 months etc?

Response: Thank you for your valuable input! We have now included in that sentence the time of the start and the length of the interventions right after the sentence you are referring to in section Discussion – Previous research (lines 315-317): “The interventions included in that review started pre-surgery or between three to 102 months post-surgery and the lengths of the interventions were between six and 12 weeks long [15].”

Page 13 line 294 – physical activity increased by how much?

Response: Thank you for this input! We have now added that information in a paragraph right after the end of that sentence you are referring to (lines 326-327): “…(counselling and pedometers = 6 787 steps/day, pedometers only = 5 325 steps/day and standard care = 5253 steps/day) [34].”

Page 14 line 314 – attendance at at least one session, was this one session the physical activity session or not? If not why not? If not how many of those included attended that session?

If they did not attend the physical activity session this is likely why you didn’t see a change and previous research in this population reports exercise education alone is insufficient for increasing physical activity.

Response: Thank you for this comment! This comment is in line with a previous comment from you above, about how many attended the session about physical activity. When “at least one session” is mentioned in the manuscript, it is at least one of any of the four sessions in the intervention. We have added, after input from you in the comment above, information and per protocol analysis on how many participants attended the session about physical activity and the results from the per protocol analysis. We therefore refer to our answer on that question above where we have stated all the changes we made to the manuscript according to your comment. We have now, after this comment, also clarified that we mean at least any of the four sessions (line 347): “… 73% attended at least one session (any of the four sessions)…”

The same addition (any of the four sessions) have been added in the section Results – Descriptive analysis (line 225): “…had attended at least one group session (any of the four sessions),…”.

Limitation – may not have attended the physical activity session

Response: Thank you for this valuable input! You are right, we have not mentioned that as a limitation. We have now added that to the section Discussion – Limitations (lines 402-404): “Fifth, only 61.6% of the participants in the intervention group attended the session about physical activity, which might have had impact on the results.”

You have mentioned you were likely not powered, have you looked at your data to see how many individuals you would have needed to be powered?

Response: Thank you for this comment! However, no we have not looked at that data, as we calculated the power of the intervention according to HRQoL (which is part of another yet not published article) as the initial aim of this intervention was to try to maintain/prevent the decline in HRQoL post-surgery. 

Fig 1 – include amount of individuals who attended the physical activity session

Response: Thank you for this input! We have now included this information in the figure caption of Figure 1 (lines 120-122): “Of the included participants in the two-year follow-up belonging to the intervention group, 61 of them (61.6%) had attended the session about physical activity.”

Reviewer #3:

Two-year follow-up results on patients activity levels were summarized from a two-arm randomized controlled clinical trial which aimed to increase quality of life following Roux-en-Y gastric bypass surgery through dissonance-based group intervention. At two-years follow-up, no significant differences in physical activity levels were observed between the intervention and control arms.

Minor revisions:

1- Cohen’s d is valid when the distribution of the data is normal. Indicate if the normality assumption was met.

2- Line 172: Technically the Kruskal-Wallis H test is used for comparing ranks rather than means for data that is not normally distributed. Additionally, non-normally distributed data is generally summarized using median, first and third quartiles.

Response: Thank you for these comments! As these two comments are related, we will answer them together. We have now added a paragraph in the section Method – Statistical analysis on information about normality assumptions and why we chose to use Kruskal-Wallis H test and Cohen’s D (lines 185-189): “Normal distribution was evaluated graphically and tested with Shapiro-Wilk test. The variables were approximately normally distributed, however not perfectly. We therefore present the results as means and use Cohen’s d to quantify intervention effect sizes [31], but because of the lack of complete normality, we also used Kruskal-Wallis H test to test for differences in medians between the groups.”

We also added information under Table 2 (lines 254-256): “…p-value for the difference in medians between the two groups (Kruskal-Wallis H test) at pre-RYGB, one- and two-years post-RYGB surgery. Effect sizes at two years measured with Cohen’s d (95 % CI). *Normal distribution assumption rejected according to Shapiro-Wilk test (p<0.05).”

3- Line 173 states, “Chi-square test was used for dichotomous variables.” Provide further clarity by indicating the comparisons which are being tested by the chi-square test.

Response: Thank you for the opportunity to clarify this! Variables that were dichotomous were for example “smokers or not”, “university level or not” and “meeting physical activity recommendations or not”. We have now included thin in the section Method – Statistical analysis (lines 191-192): “Chi-square test was used for dichotomous variables, for example if a participant smoked or not, had university level of education or not, and meeting the physical activity recommendations or not.”

4- Line 177 would be clarified by stating, “Sensitivity analysis was conducted for the participants with valid accelerometer measurements from all three time points, conducted with a regression analysis adjusted for baseline measures and presented as differences of the means between the groups at one- and two-years follow-up.”

Response: Thank you for letting us clarify this sentence! We have now changed it according to your suggestion (line 198-201): “Sensitivity analysis was conducted for the participants with valid accelerometer measurements from all three time points, conducted with a regression analysis adjusted for pre-RYGB measures and presented as differences of the means between the groups at one- and two-years follow-up.”

5- Line 180: Again replace “calculated” with “conducted.”

Response: Thank you for letting us improve the sentence! We have now changed it according to your suggestion (line 202): “Additionally, a sensitivity analysis was conducted for all participants who had ≥5 valid days…”

6- Table 2: Clarify the statistical methods used to calculate the p-values in Table 2.

Response: Thank you for the comment! We have now clarified this under Table 2 and added how we calculated the p-value (line 254): “p-value for the difference in medians between the two groups (Kruskal-Wallis H test) at pre-RYGB, one- and two-years post-RYGB surgery.”

We have also added the same information under S2 Appendix and S3 Appendix.

7- Line 232: Replace “numbers” with “frequency” to improve clarity.

Response: Thank you for the comment! We have now changed the sentence according to your suggestion in the text under Table 2 (line 254): “…or frequency (percent) for each subscale…”

We have also added that same information under S2 Appendix and S3 Appendix.

8- Clarify if SE represents standard error or standard error of the mean.

Response: Thank you for the comment! We do mean Standard Error of the mean.

---

## [Editor Report · Decision Letter 2]

21 Jul 2021

Physical activity in women attending a dissonance-based intervention after Roux-en-Y Gastric Bypass: A 2-year follow-up of a randomized controlled trial

PONE-D-20-01663R2

Dear Dr. Berglind,

We’re pleased to inform you that your manuscript has been judged scientifically suitable for publication and will be formally accepted for publication once it meets all outstanding technical requirements.

Kind regards,

Dario Ummarino, Ph.D.

Senior Editor

PLOS ONE
---

## [Editor Report · Acceptance letter]

12 Oct 2021

PONE-D-20-01663R2 

Physical activity in women attending a dissonance-based intervention after Roux-en-Y Gastric Bypass: A 2-year follow-up of a randomized controlled trial 

Dear Dr. Berglind:

I'm pleased to inform you that your manuscript has been deemed suitable for publication in PLOS ONE. Congratulations! Your manuscript is now with our production department. 

Kind regards, 

on behalf of

Dr. Dario Ummarino 

Staff Editor

PLOS ONE